# Native Rhizobial Inoculation Improves Tomato Yield and Nutrient Uptake While Mitigating Heavy Metal Accumulation in a Conventional Farming System

**DOI:** 10.3390/microorganisms13081904

**Published:** 2025-08-15

**Authors:** Luis Alberto Manzano-Gómez, Clara Ivette Rincón-Molina, Esperanza Martínez-Romero, Simón Samuel Stopol-Martínez, Amado Santos-Santiago, Juan José Villalobos-Maldonado, Víctor Manuel Ruíz-Valdiviezo, Reiner Rincón-Rosales

**Affiliations:** 1Laboratorio de Ecología Genómica y Agricultura Regenerativa, Tecnológico Nacional de Mexico, Instituto Tecnológico de Tuxtla Gutiérrez, Tuxtla Gutiérrez 29050, Chiapas, Mexico; contacto@3rbiotec.com (L.A.M.-G.); juan.vm@tuxtla.tecnm.mx (J.J.V.-M.); victor.rv@tuxtla.tecnm.mx (V.M.R.-V.); 2Departamento de Investigación y Desarrollo, 3R Biotec, Tuxtla Gutiérrez 29000, Chiapas, Mexico; 3Centro de Ciencias Genómicas, Universidad Nacional Autónoma de México, Cuernavaca 62210, Morelos, Mexico; esperanzaeriksson@yahoo.com.mx; 4Departamento de Bioinformática, Gensoil, Fountain Valley, CA 92708, USA; contacto@gensoil.net; 5Tecnológico Nacional de México, Instituto Tecnológico de Tlaxiaco, Oaxaca 69800, Oaxaca, Mexico; dir_tlaxiaco@tecnm.mx

**Keywords:** plant probiotics, conventional farming, diazotrophic *Rhizobium*, Fe and Cu accumulation, native *Sinorhizobium*, nutrient content in plant tissue, soil bacterial community, tomato yield

## Abstract

Enhancing crop productivity through biological strategies is critical for agriculture, particularly under conventional farming systems heavily reliant on chemical inputs. Plant probiotic bacteria offer promising alternatives by promoting plant growth and yield. This is the first field study to assess the effects of biofertilization with native rhizobial strains *Rhizobium* sp. ACO-34A, *Sinorhizobium mexicanum* ITTG-R7^T^, and *S. chiapasense* ITTG-S70^T^ on *Solanum lycopersicum* (tomato) cultivated under conventional farming conditions. Key parameters assessed include plant performance (plant height, plant stem width, plant dry weight, and chlorophyll content), fruit yield (fruits per plant, fruit height, fruit width, fruit weight, and estimated fruit volume), and macronutrient and micronutrient contents in plant tissue. Additionally, rhizospere bacterial communities were characterized through 16S rRNA amplicon sequencing to evaluate alpha and beta diversity. Inoculation with ITTG-R7^T^ significantly improved plant height, stem width, and plant dry weight, while ITTG-S70^T^ enhanced stem width and chlorophyll content. ACO-34A inoculation notably increased fruit number, size, and yield parameters. Moreover, inoculated plants exhibited reduced Fe and Cu accumulation compared to non-inoculated controls. Metagenomic analyses indicated that rhizobial inoculation did not significantly disrupt the native rhizosphere bacterial community. These findings highlight the potential of rhizobial strains as effective plant probiotics that enhance tomato productivity while preserving microbial community structure, supporting the integration of microbial biofertilizers into conventional farming systems.

## 1. Introduction

Chemical fertilizers have played a key role in agriculture, but their excessive and constant application has contributed to reduce soil health [1]. Farming practices lacking soil data often result in excessive agrochemical use to increase yields but without considering a holistic approach that integrates the physical, chemical, and biological properties of the soil. Although this is not representative of all farmers, particularly those with better agronomic guidance, this approach can result in nutrient imbalances, soil degradation, and environmental contamination, contributing to water and air pollution as well as biodiversity loss [2]. For instance, on average, 50% of applied nitrogen and 0.01–1% of applied phosphorous are effectively assimilated by plants, while the remaining are lost through volatilization (~30%), leaching (~40%), and denitrification (~5% to 30%) [3]. Also, overfertilization can lead to heavy metal accumulation in plant tissues, particularly iron (Fe), which may reduce vegetable crop yields [4].

Given these challenges, harnessing microbial-based technologies, such as elite plant probiotic bacteria and plant growth-promoting bacteria (PPB/PGPB), represents a promising alternative to enhance nutrient use efficiency, reduce the environmental footprint of agricultural inputs, and improve crop yields [5].

Biofertilizers formulated with PPB, particularly those based on beneficial rhizobial strains, have gained attention as sustainable alternatives to enhance crop yields [6]. PPB are defined as live microorganisms that, when administered in specific amounts, confer health benefits to the host. Many PPB are classified as the PGPB group, as they colonize plant roots, enhance nutrient availability, and improve resistance to abiotic and biotic stresses [7,8]. However, a common misconception in biofertilizer development is the assumption that any microorganism exhibiting PGP traits, such as phosphorous, potassium, and zinc solubilization, nitrogen fixation, or siderophore production, can be used as a biofertilizer, when in fact not all PGPB are suitable for biofertilizer formulation due to the potential biological risks they may pose to ecosystems, animals, and humans (One Health approach) [9,10]. Several genera, including *Burkholderia*, *Pseudomonas*, *Serratia*, *Acinetobacter*, *Enterobacter*, *Klebsiella*, *Ralstonia*, and *Bacillus*, are associated with respiratory and gastrointestinal infections in humans [9].

Although *Rhizobium* species are traditionally associated with leguminous crops, studies have demonstrated that some rhizobial strains, such as *Rhizobium leguminosarum* PETP01 and TVP08, *R. laguerreae* PEPV40, *R. leguminosarum* PEPV16 [10,11,12,13], *Sinorhizobium mexicanum* ITTG-R7^T^, *Sinorhizobium chiapasense* ITTG-S70^T^, and *Rhizobium* sp. ACO-34A [6,14,15,16], can enhance the growth and productivity of non-leguminous crops such as lettuce, carrot, spinach, strawberry, guava, and agave by improving nutrient uptake, root development, and stress tolerance. Gen-Jiménez et al. [17] reported that native *Rhizobium calliandrae* LBP2-1, *R. mayense* NSJP1-1, and *R. jaguaris* SJP1-2 enhanced tomato productivity in microcosm experiments through multiple PGP mechanisms, including cellulose production, biofilm formation, and root colonization.

Despite the increasing interest in biofertilizers, most studies focus on alternative farming systems, such as organic agriculture. However, limited research has evaluated the integration of *Rhizobium* and *Sinorhizobium* as PPB within conventional farming systems, where synthetic fertilizers and agrochemicals remain dominant. Understanding their compatibility in these systems is essential for their large-scale adoption. Le Campion et al. [18] define conventional farming as an agricultural system characterized by frequent applications of synthetic fertilizers, fungicides, insecticides, herbicides, tillage, and irrigation.

Tomato (*Solanum lycopersicum*) is one of the most widely cultivated vegetable crops worldwide due to its economic importance; however, its production is heavily dependent on chemical fertilizers and intensive agricultural inputs [19,20]. In 2022, major tomato-producing regions including the United States (California), Italy, and China represented the largest production regions (https://www.morningstarco.com/2023-post-season-global-tomato-crop-update/) (accessed on 9 September 2024).

The aims of this study were to evaluate the effects of native strains *Sinorhizobium mexicanum* ITTG-R7^T^, *Sinorhizobium chiapasense* ITTG-S70^T^, and *Rhizobium* sp. ACO-34A on plant growth, fruit yield, nutrient content in plant tissue, and the rhizosphere bacterial community structure of tomato (*Solanum lycopersicum*) under real-field conditions. To achieve this, we integrated metagenomics, bioinformatics, and advanced statistical techniques, including principal component analysis (PCA), Linear Discriminant Analysis Effect Size (LEfSe), and Redundancy Analysis (RDA). Our findings aim to support the development and implementation of native *Rhizobium*-based biofertilizers as viable alternatives to enhance tomato crop productivity in conventional farming.

## 2. Materials and Methods

### 2.1. Field Site Description

This study was conducted at the “Santa Rosa” production site, which encompasses a total area of 10 ha (16°43′23.0″ N 93°16′42.6″ W), located in Chiapas, México (16°47′59.0″ N 93°16′21.4″ W) (Figure 1), at an altitude of 1046 masl. The soil at Santa Rosa is classified as Leptosols according to the World Reference Base (https://soilgrids.org/ (accessed on 3 March 2025)). Leptosols are very shallow soils with minimal development, typically formed on hard rock or highly calcareous materials and are considered azonal soils in the pre-Soil Taxonomy classification of the USDA [21]. The climate is classified as warm sub-humid (AwO) with summer rains, an average annual temperature of 23 °C, and an average annual rainfall of 1450 mm. The main cultivated crops at Santa Rosa include tomato (*Solanum lycopersicum*), habanero pepper (*Capsicum chinense*), and cucumber (*Cucumis sativus*) under a 100% irrigation regime in two production cycles: spring–summer and autumn–winter. Regarding agricultural residue management, crop residues are commonly incinerated on site. In terms of soil agronomic management, conventional tillage is applied, consisting of one pass of a disc plow (25 cm depth) and two passes of a harrow (15 cm depth). Fertilization relies on chemical fertilizers, applied empirically without prior soil fertility analysis. Farmers typically use ammonium sulfate as a N source, monoammonium phosphate as a P source, and potassium sulfate as a K source. Weed control is performed using 1,1′-dimethyl-4,4′-bipyridyl dichloride, a fast-acting, non-selective herbicide for the control of broadleaf and grass weeds, and pest and disease control is managed through agrochemical applications.

### 2.2. Sampling Site and Field Experiment Establishment

The experiment was conducted on a 3400 m^2^ open field (36.8 m wide × 92.4 m long) during the spring–summer season (2023–2024) (Figure 1d). Prior to the experiment’s establishment, an initial bulk soil sample (BS_I) was collected for physicochemical characterization on 22 December 2023. At the conclusion of the experiment (19 May 2024), a second bulk soil sample (BS_F) was collected. Soil samples were taken from five randomly selected points across the plot, each in triplicate, following the methodology described by Manzano-Gómez et al. [22]. Subsamples were obtained by coring the top 15 cm of soil, which were then combined to form a composite sample. The composite soil samples were stored in Whirl-Pak^®^ bags (Nasco Sampling LLC, Madison, WI, USA) and transported in an icebox to the Genomic Ecology and Regenerative Agriculture laboratory at the National Technological of Mexico (TecNM) campus Tuxtla Gutiérrez for subsequent analysis. The bulk soil samples (BS_I and BS_F) were analyzed for properties such as texture, saturation point (SP; %), water-holding capacity (WHC; %), permanent wilting point (PWP; %), hydraulic conductivity (HC; cm h^−1^), bulk density (BD; g cm^−3^), pH, total carbonates (TC; %), electrical conductivity (EC; dS m^−1^), soil organic matter (SOM; %), and cation exchange capacity (CEC; me 100 g^−1^). Additionally, macronutrient and micronutrient concentrations were determined: phosphorus (P; ppm), potassium (K; ppm), calcium (Ca; ppm), magnesium (Mg; ppm), sodium (Na; ppm), iron (Fe; ppm), zinc (Zn; ppm), manganese (Mn; ppm), copper (Cu; ppm), boron (B; ppm), sulfur (S; ppm), and nitrates (NO_3_; ppm). These analyses were conducted according to the Official Mexican Standard NOM-021-SEMARNAT-2000 [6,23].

Following bulk soil characterization, twenty-six ridges were established with each ridge spaced 1.0 m apart. After ridge formation, a micro-sprinkler irrigation system (1.1 L h^−1^ flow rate) (Irritec^®^, Capo d’Orlando, Sicily) was installed to provide targeted irrigation at the plant base. Finally, plastic mulch was applied to each ridge (Figure 2). The use of plastic mulch is a common practice in horticultural crops, as it helps retain soil moisture by reducing evaporation from the soil surface [24].

### 2.3. Experimental Design

A completely randomized design (CRD) was employed, with four treatments and each replicated three times. A CRD was selected as it is a statistical technique that reduces experimental error variance, thereby improving the precision of response variables [25]. Although a total of 26 ridges were initially established by the farmer, 6 ridges (three on each side) were excluded to minimize border effects, leaving 20 ridges available. From these, 12 ridges were assigned to the experimental design, as shown in Figure 3a.

The evaluated treatments were as follows: T_1_: non-inoculated (control), T_2_: *Rhizobium* sp. ACO-34A [10 mL plant^−1^ at 10^6^ CFU mL^−1^], T_3_: *S. mexicanum* ITTG-R7^T^ [10 mL plant^−1^ at 10^6^ CFU mL^−1^], and T_4_: *S. chiapasense* ITTG-S70^T^ [10 mL plant^−1^ at 10^6^ CFU mL^−1^]. It is important to note that all 26 ridges, including those receiving biological treatments (T_2_, T_3_, and T_4_) were chemically fertilized according to farmer management, as the experimental plot was continuously managed under conventional farming. Chemical fertilization consisted of the application of 440 kg ha^−1^ (NH_4_)_2_SO_4_ [ammonium sulfate], 60 kg ha^−1^ NH_4_H_2_PO_4_ [monoammonium phosphate], 180 kg ha^−1^ K_2_SO_4_ [potassium sulfate], 150 kg ha^−1^ MgSO_4_ [magnesium sulfate], 10 kg ha^−1^ MnSO_4_ [manganese sulfate], 3.5 kg ha^−1^ BO_2_ [boron oxide], 10 kg ha^−1^ ZnSO_4_ [zinc sulfate], and 1 kg ha^−1^ Fe chelate (Fe was applied each week via fertigation within two and a half months after microgreens transplantation).

### 2.4. Bacterial Strains

The native elite strains *Sinorhizobium mexicanum* ITTG-R7^T^ (DQ411930, type strain), Sinorhizobium chiapasense ITTG-S70^T^ (EF457949, type strain), and *Rhizobium* sp. ACO-34A (KM349967) employed in this study are known for their plant probiotic and plant growth-promoting traits. The *Sinorhizobium* strains are novel symbiotic nitrogen-fixing bacterial species isolated from the shrubby legume *Acaciella angustissima* nodule [26], characterized by its high capacity to solubilize phosphate and synthesize auxins [6], while *Rhizobium* sp. is a free-living diazotrophic strain isolated from the rhizosphere of *Agave americana* L. [14] coupled with robust PGP traits, such as nitrogen fixation, phosphate solubilization, siderophore production, and indole-3-acetic acid (IAA) production, which are crucial for enhancing nutrient availability [15,16]. These strains do not exhibit virulence factors for human, animals, or plants and are not listed in the Global Catalogue of Pathogens [27]. They are maintained in the culture collection of the Genomic Ecology and Regenerative Agriculture Laboratory at TecNM, Tuxtla Gutiérrez, Chiapas.

### 2.5. Bacterial Inoculation Trial in Tomato Crop

The ACO-34A, ITTG-R7^T^, and ITTG-S70^T^ strains were grown in YEM + Ca^2+^ medium containing the following per liter: 10 g of mannitol, 0.5 g of K_2_HPO_4_, 0.2 g of MgSO_4_, 0.1 g of NaCl, 0.3 g of CaCO_3_, and 3.0 g of yeast extract al pH 6.8. We followed the methodology described by Manzano-Gómez et al. [28]. The final concentration was adjusted to 10^6^ CFU mL^−1^, a threshold considered sufficient to elicit a positive inoculation response [29]. For the non-inoculated control treatment, a total of 208 tomato microgreens (*Solanum lycopersicum* var. Bola) were transplanted per ridge. For each biological treatment (T_2_, T_3_, and T_4_), 208 tomato microgreens per ridge were inoculated with 10 mL plant^−1^, applied directly to the plant base before manual transplantation. Plant spacing was maintained at 40 cm. Two additional applications of 10 mL plant^−1^ of bacterial suspension (10^6^ CFU mL^−1^) were applied directly at the plant base at 25 and 40 days after transplantation (DAT). The inoculation of the tomato microgreens and their transplant was carried out on 30 December 2023 (Figure 3b).

### 2.6. Assessment of Morphological Traits, Fruit Yield, and Nutritional Composition in Plant Tissue

All measurements and sample collections were performed on 14 March 2024. The following morphometric parameters were measured to determine the effect of bacterial inoculation on tomato plants: plant height (cm), stem diameter (mm), and plant dry weight (g) (only the phyllosphere, excluding fruits). The relative chlorophyll content was determined using the SPAD (Soil Plant Analysis Development) index with a chlorophyll meter.

To assess fruit yield, the following parameters were measured: fruits per plant (evaluated in one hundred and twenty plants per treatment), fruit height (cm), fruit width (cm), and fruit weight (g) (measured from sixty fruits per treatment). Only ripe, market-ready tomatoes (excluding green fruits) were considered. The estimated fruit volume (EFV) (cm^3^) was calculated using the following formula:(1)$$EFV\;=\;\frac{\pi }{6}\times F_{h}\times {F_{w}}^{2}$$
where *F_h_* is fruit height and *F_w_* is fruit width.

To analyze macronutrient and micronutrient content in tomato plant tissue dry weight (including stem, branches, and leaves), three composite samples were obtained for each treatment from twenty plant subsamples per ridge. Nitrogen was determined by using the Dumas method (AOAC, 2005) with an elemental analyzer FlashEA 1112^®^ (ThermoFisher Scientific, Waltham, MA, USA) which involves high-temperature oxidation of organic matter and thermal conductivity detection of the resulting nitrogen gas. The content of other elements, such as phosphorus, potassium, calcium, magnesium, sulfur, sodium, iron, zinc, manganese, copper, boron, nickel, and molybdenum, was determined according to AOAC method 2006.03, which consists of microwave multi-element acid digestion followed by quantification using inductively coupled plasma optical emission spectrometry (ICP-OES). The accumulated nutrient content in plant tissue was calculated by multiplying the nutrient concentration by the plant dry weight.

### 2.7. Molecular Analysis of Bacterial Communities

The structure and composition of bacterial communities from BS_I, BS_F and rhizospheric soil samples from each treatment were analyzed using 16S amplicon metagenomics. DNA from each sample was extracted using the ZymoBiomics^®^ DNA Miniprep kit (Zymo Research Corporation, CA, USA). The purity and concentration were determined using a NanoDrop^®^ One (ThermoFischer Scientific, Waltham, MA, USA). DNA samples were sent to the Macrogen DNA sequencing service (Seoul, Republic of Korea) for amplification. The V3-V4 hypervariable region of the 16S *r*RNA gene was amplified using primers Bakt_341F (CCTACGGGNGGCWGCAG) and Bakt_805R (GACTACHVGGGTATTAATCC) on the Illumina^®^ Miseq 2000 platform (Illumina, Inc., San Diego, CA, USA) with a paired-end sequencing configuration (2 × 300 bp). Sequencing data quality control was performed using FASTQC v0.11.9 software [30]. Bioinformatics filtering and alignment were conducted using the Phyloseq v1.41.1 [31] and DADA2 v1.16 [32] packages in the open-source software RStudio v4.1.1 [33]. Downstream bioinformatics for bacterial community analysis was conducted using a cluster of algorithms; HMMER’s hmmsearch algorithm was used to remove non-specific amplicons that do not encode 16S rRNA [34], DUDE-seq was used for denoising sequences [35], the UCHIME algorithm was used for chimera detection [36], the UCLUST algorithm was used for clustering and extraction of non-redundant reads, and the USEARCH algorithm was used for taxonomic identification [37] against the PKSSU4.0 database [38]. Alpha diversity indices, including ACE, Chao1, Shannon, and Non-parametric Shannon (NP Shannon), were then calculated, and relative abundance profiling across the samples was conducted using ggplot2 v3.5.1 [39]. Beta diversity was assessed using principal coordinated analysis (PCoA) with the Bray–Curtis dissimilarity method. Statistical significance was determined using permutational multivariate analysis of variance (PERMANOVA) [17]. Alpha and beta diversity plots were generated using ImageGP v2.0 [40]. Functional predictions of the bacterial community based on 16S rRNA sequences were conducted using the PICRUSt v2.6.1 algorithm [41]. The raw sequence data obtained in this study were deposited in the NCBI Sequence Read Archive (SRA) under accession number PRJNA1004673.

### 2.8. Statistical Analysis

Soil physicochemical data were analyzed using paired samples Student’s *t*-test (α = 0.05), while the data from inoculation trial were analyzed using ANOVA at a significance level of α = 0.05. Mean comparisons were performed using Tukey’s test (*p* < 0.05) with the agricolae package v1.3-7 [42] in RStudio v4.1.1 [33]. The coefficient of variation (CV) was estimated for plant morphometric characteristics and fruit yield using the following formula:(2)$$CV=\frac{S}{\bar{Χ}}$$
where $\bar{Χ}$ is the trait grand mean estimated from the experiment and $S\;=\;\sqrt{S^{2}}$, with *S*^2^ representing the residual variance in the experiment. CV is commonly applied to present the variation in agricultural traits and can be used to compare the variability in different traits [43]. Principal component analysis (PCA) was conducted using the factoextra v1.0.7 [44] and Corrplot v0.95 [45] packages. Correlation analysis (CA) was conducted using the Pearson (*r*) method to explore the association between two or more variables. Additionally, the Linear Discriminant Analysis Effect Size (LEfSe) method was implemented to identify functional biomarkers across treatments [46]. Redundancy analysis (RDA) was performed to explore the relationships between bacterial community and soil physicochemical properties in relation to tomato crop yield [47].

## 3. Results

### 3.1. Bulk Soil Physicochemical Characteristics

The initial bulk soil (BS_I) was classified as moderately alkaline (pH: 7.96 ± 0.23) and highly calcareous due to its high total carbonate content (TC: 26.30% ± 9.01). It exhibited a clay texture with a high water-holding capacity (WHC: 42.17% ± 2.16) and a high level of soil organic matter (SOM: 5.39 ± 0.16). Regarding macronutrient availability, BS_I showed a moderate supply of inorganic nitrogen (NO_3_: 31.93 ± 1.99 ppm) and a high supply of available phosphorus (26.10 ± 15.86 ppm). In terms of micronutrients, it had a moderately low zinc content (1.04 ± 0.01 ppm), was deficient in manganese (2.85 ± 0.15 ppm), and had a high level of copper (28.97 ± 0.78 ppm). Additional bulk soil characteristics are presented in Table 1.

At the end of the experiment (BS_F), an increase in pH (8.67 ± 0.34) and a reduction in SOM content (5.01% ± 0.15) were observed, although these changes were not statistically significant (*p* > 0.05). In contrast, a significant change in hydraulic conductivity (HC) was observed. Reductions in water-holding capacity (WHC: 33.84% ± 2.44), saturation point (SP: 63.36% ± 4.83), permanent wilting point (PWP: 20.47%± 1.74), and bulk density (BD: 1.13 ± 0.03 g cm^−3^) were observed, but these differences were not significant (*p* > 0.05).

Regarding macronutrients, BS_F exhibited a significant increase (*p* < 0.05) in NO_3_ (38.40 ± 0.94 ppm), K (447.89 ± 35.73 ppm), and S (26.97 ± 0.98 ppm) compared to BS_I. Additionally, an increase in Mg content (782.67 ± 34.15 ppm) and a reduction in P content (20.24 ± 0.88 ppm) were observed but without significant changes. Regarding micronutrients, the levels of Na, Zn, and B significantly increased (*p* < 0.05) in BS_F compared to BS_I, while Mn showed a slight but non-significant increase. In contrast, Fe and Cu levels significantly decreased in BS_F.

### 3.2. Impact of Inoculation on Tomato Cultivation

Maintaining or enhancing crop yields under conventional farming remains a significant challenge. To address this, we evaluated the influence of three native elite rhizobial strains, ITTG-R7^T^, ITTG-S70^T^, and ACO-34A, on plant morphometric traits, fruit yield, and nutrient content in plant tissue.

Significant differences (*p* < 0.05) were observed in morphometric traits among treatments (Table 2). Plant height in T_3_ (ITTG-R7^T^) (119.33 ± 3.06 cm) and T_1_ (non-inoculated) (117.33 ± 3.79 cm) was statistically similar but significantly higher compared to the other treatments. For stem width, T_3_ (16.12 ± 3.10 mm) and T_4_ (15.41 ± 3.51 mm) showed significant increases (*p* < 0.05) compared to T_1_ (12.45 ± 1.91 mm). In terms of plant dry weight, representing the biomass from stems, branches, and leaves, T_3_ (306.67 ± 15.28 g) presented the highest biomass accumulation, followed by T_1_ (246.67 ± 68.25 g). The lowest dry weights were recorded in T_2_ (ACO-34A) (170.00 ± 52.92 g) and T_4_ (ITTG-S70^T^) (135.00 ± 13.23 g) (*p* < 0.05). Regarding chlorophyll content, measured via the SPAD index, significant differences were observed. Plants inoculated with T_4_ showed the highest chlorophyll content (43.19 ± 2.39 SPAD), while the lowest values were observed in T_1_ (37.09 ± 3.63 SPAD).

### 3.3. Fruit Yield in Inoculated Plants

Plants inoculated with ACO-34A exhibited significant improvements (*p* < 0.05) across all measured parameters, including the number of fruits per plant, fruit height, fruit width, fruit weight, and estimated fruit volume (EFV) (Table 3). The effect on the size and weight of tomatoes treated with ACO-34A compared to the non-inoculated treatment (T_1_) was notable (Appendix A). Also, T_3_ (ITTG-R7^T^) and T_4_ (ITTG-S70^T^) significantly increased the number of fruits per plant in comparison with T_1_ (non-inoculated plants).

### 3.4. Nutritional Content in Plant Tissue

Inoculation with the ITTG-R7^T^ strain (T_3_) showed a significantly higher accumulation of total nitrogen (N), phosphorous (P), and potassium (K) in dried plant tissue. In contrast, non-inoculated plants (T_1_) showed significantly higher levels (*p* < 0.05) of iron (Fe), copper (Cu), nickel (Ni), and molybdenum (Mo) compared to biofertilized treatments. The accumulation of other macronutrients and micronutrients across treatments is presented in Table 4.

Principal component analysis (PCA) was conducted to explore associations among morphometric characteristics, fruit yield, and nutritional content in inoculated plants with native rhizobial strains. The first two principal components explained 81.4% of the total variance. The ACO-34A treatment (T_2_) was primarily associated with fruit yield parameters, including estimated fruit volume (EFV), fruit height, fruit width, fruit weight, and fruits per plant. In contrast, the ITTG-R7 ^T^ treatment (T_3_) clustered closely with variables related to nutrient accumulation in plant tissue, particularly macronutrients (N, P, K, Ca, Mg, and S) and micronutrients (Na, Zn, Mn, and B), alongside positive trends observed in plant dry weight and plant height. The non-inoculated plants (T_1_) grouped near variables associated with plant height and higher concentrations of Fe, Cu, Ni, and Mo (Figure 4).

Interestingly, we observed a negative association between the accumulation of metals in plant tissue (Fe, Cu, Ni, and Mo) and plant morphometric characteristics, particularly with traits like plant stem width and chlorophyll, and all fruit yield traits (fruits per plant, fruit height, fruit width, fruit weight, and estimated fruit volume (EFV)). To confirm these findings, Pearson’s correlation (*r*) analysis was performed. Fruit weight showed a significant negative correlation with iron (Fe) (*r*: −0.6821, *p*-value: 0.0146) and Cu accumulation (*r*: −0.6257, *p*-value: 0.0296). Similarly, the estimated fruit volume (EFV) was negatively correlated with Fe (*r*: −0.7526, *p*-value: 0.0047) and Cu (*r*: −0.6974, *p*-value: 0.0116) accumulation. Although fruits per plant were also negatively correlated with metal accumulations in plant tissue, no significant differences were determined. Additional correlations are presented in Appendix A.

### 3.5. Bacterial Community Structure

Using metagenomic sequencing of 16S rRNA gene amplicons, a total of 18 raw sequences were obtained from bulk soil (BS_I and BS_F) and rhizosphere soil samples of tomato plants (T_1_, T_2_, T_3_, and T_4_). After filtering, 847,554 valid reads were obtained, with an average of 47,086 reads per sample. The valid sequences were grouped into Operational Taxonomic Units (OTUs) with 97% similarity, averaging 3967 OTUs per sample.

The study of the bacterial community in bulk soil revealed changes in the relative abundance of the phyla. In the initial bulk soil (BS_I), the most abundant phylum was Actinobacteria (36%), followed by Acidobacteria (24%) and Proteobacteria (22%). In contrast, in the final bulk soil (BS_F), the relative abundance of Acidobacteria increased to 39% and Actinobacteria decreased to 16%, while Proteobacteria remained stable at 22% (Figure 5a).

In the rhizosphere, an enrichment in the relative abundance of Proteobacteria was observed across all treatments. However, this enrichment was more pronounced in plants inoculated with the ITTG-S70^T^ strain, compared to the non-inoculated plants (T_1_) (Figure 5a). Although differences in community structure were observed, particularly in the phyla Proteobacteria (*p*-value: 0.3274), Actinobacteria (*p*-value: 0.3323), and Acidobacteria (*p*-value: 0.3274), these differences were not statistically significant.

### 3.6. Diversity and Species Richness of Bacterial Community

The species richness and diversity of the bacterial community were estimated using the ACE and Chao1 indices (for richness) and the Shannon and NP Shannon indices (for diversity) in both bulk soil (BS_I and BS_F) and rhizosphere samples (T_1_, T_2_, T_3_, and T_4_). In the initial bulk soil (BS_I), the indices were as follows: ACE (4725.07 ± 535.39), Chao1 (4435.7 ± 525.27), Shannon (6.96 ± 0.14), and NP Shannon (7.09 ± 0.13). In the final bulk soil (BS_F), a decrease in both species richness and diversity indices was observed. However, these differences were not statistically significant (*p* > 0.05) (Figure 5b). Similarly, tomato plants inoculated with the ACO-34A (T_2_), ITTG-R7^T^ (T_3_), and ITTG-S70^T^ (T_4_) strains did not show statistically significant changes in species richness and diversity compared to the non-inoculated plants (T_1_) based on the ACE, Chao1, Shannon, and NP Shannon indices (Figure 5b). Additionally, orrelation analysis was performed between the diversity indices (ACE, Chao, Shannon, and NP Shannon) and soil physicochemical properties. Although a positive correlation with SOM and a negative correlation with pH were identified, these correlations were not statistically significant (*p* > 0.05).

### 3.7. Core Soil Bacterial Community

Six phyla (Actinobacteria, Acidobacteria, Proteobacteria, Planctomycetes, Chloroflexi, Verrucomicrobia) were identified with a relative abundance greater than 1% and were consistently present across both bulk soil and rhizospheric soil samples (Figure 5c).

### 3.8. Beta Diversity Analysis and Functional Biomarkers

We conducted the beta diversity analysis using the Bray–Curtis index on the bacterial communities of bulk soil and rhizospheric soil from inoculated plants at the genus level (Figure 5d). The results showed differences between the bacterial communities of plants inoculated with ACO-34A, ITTG-R7^T^, and ITTG-S70^T^ strains and the controls treated with chemical fertilizers, as well as between the BS_I and BS_F. Although the PERMANOVA test yielded a *p*-value = 0.08, which did not reach statistical significance, there was a trend suggesting differences between treatments. This trend was further supported by a homogeneity test assessing data dispersion for each sample, confirming that the observed differences are attributable to the applied treatments rather than internal variability among the samples (*p*-value: 0.786).

Regarding functional biomarkers, the LEfSe analysis revealed that plants inoculated with ACO-34A, ITTG-R7^T^, and ITTG-S70^T^ strains enriched the abundance of orthologous genes in the rhizosphere of tomato plants. Among the identified orthologs, oligo-1,6-glucosidase (K01182) and pyrimidine/purine-5′-nucleotide nucleosidase (K06966) were enriched with the ACO-34A strain. Meanwhile, the treatment with ITTG-R7^T^ enriched the ortholog K02840, related to UDP-D-galactose:(glucosyl) LPS alpha-1,6-D-galactosyltransferase.

Finally, the redundancy analysis (RDA) did not show a statistically significant association (pseudo-F value: 0.8273, *p*-value: 0.5974) between soil properties, the associated bacterial community, fruit yield, morphometric characteristics, and the nutritional parameters of tomato plants over the six-month experiment.

## 4. Discussion

This study is the first to report the application of elite rhizobial strains *S. mexicanum* ITTG-R7^T^, *S. chiapasense* ITTG-S70^T^, and *Rhizobium* sp. ACO-34A in open-field tomato cultivation under conventional farming systems.

Inoculation with *S. mexicanum* ITTG-R7^T^ significantly increased plant stem width and plant dry weight compared to non-inoculated plants. These positive effects observed with the inoculation of ITTG-R7^T^ highlight the potential of this native strain to enhance structural development in tomato plants. Likewise, *S. chiapasense* ITTG-S70^T^ not only promoted thicker stems but also resulted in the highest relative chlorophyll content among all treatments, suggesting that this strain may improve photosynthetic efficiency and overall plant vigor. Furthermore, inoculation with ITTG-R7^T^ led to a significant increase in the accumulation of key macronutrients (N, P, K) and micronutrients (Zn and B) in plant tissue, potentially enhancing the nutritional status and productivity of tomato crops.

The ability of rhizobial species to promote plant growth is well documented, particularly through mechanisms such as phosphorus and potassium solubilization, which enhance nutrient availability for plant uptake [6]. Both *S. mexicanum* ITTG-R7^T^ and *S. chiapasense* ITTG-S70^T^ were originally isolated from nodules of the leguminous shrub *Acaciella angustissima* and have been described as novel *Sinorhizobium* species [26,48]. Rincon-Molina et al. [6] and Maranto-Gómez et al. [16] previously evaluated the ability of these strains to solubilize phosphate, synthesize siderophores, and produce indole-3-acetic acid (IAA), confirming their potential as plant probiotic rhizobacteria. Previous studies have demonstrated that IAA-producing rhizobia can increase nutrient absorption efficiency [49].

In this study, tomato plants inoculated with the *Rhizobioum* sp. ACO-34A strain, free-living diazotrophic bacteria isolated from the rhizosphere of *Agave americana* L., had significantly increased fruit height, fruit width, and fruit weight, resulting in higher estimated fruit volumes (EFVs) compared to non-inoculated plants. Despite its limited effect on plant morphometric traits, the ACO-34A strain demonstrated its potential as a plant probiotic (PP) to enhance fruit yield and size. The mechanism by which ACO-34A influences fruit yield is likely associated with its plant growth-promoting traits, including phosphorous solubilization, siderophore production, and auxin (IAA) synthesis. Although these mechanisms were not directly evaluated in this study, previous research has documented these traits in related strains [14,15,50]. In those studies, phosphate solubilization was confirmed using dicalcium phosphate and tricalcium phosphate as phosphorus sources in yeast mannitol (YM) broth medium and quantified by calculating the phosphate solubilization index (PSI) as the ratio of halo size to colony size. Siderophore production was assessed both qualitatively and quantitatively using the Chrome Azurol S (CAS) assay. Similarly, IAA synthesis was determined via a colorimetric method using Salkowski reagent.

The divergent effects observed between *S. mexicanum* ITTG-R7^T^, *S. chiapasense* ITTG-S70^T^, and *Rhizobioum* sp. ACO-34A likely stem from their distinct ecological origins, physiological capacities, and plant interaction profiles. ITTG-R7^T^, a nodule-derived strain, may preferentially support structural and metabolic development during early growth stages through mechanisms like IAA production and phosphate solubilization [6,16]. In contrast, ACO-34A, a free-living diazotroph from the rhizosphere, may exert its effects at later stages of plant development, particularly during reproductive growth, potentially via hormone modulation and nutrient mobilization directed toward fruit formation [14]. Furthermore, the differences in colonization patterns, rhizosphere competence, and interaction with host signaling pathways may also contribute to their niche-specific impacts. Although further experimental validation is needed, these findings suggest strain-specific interactions with the host plant at different growth stages, supporting the idea of tailored inoculant strategies for distinct agronomic goals.

The efficacy of native *Rhizobium* species as PPB has also been demonstrated in other studies. For instance, microcosm experiments using *Rhizobium calliandrae* LBP2-1^T^ (JX855162), *Rhizobium mayense* NSJP1-1^T^ (JX855172), and *Rhizobium jaguaris* SJP1-2^T^ (JX855169) have shown improvements in tomato fruit size and quality [17].

Maximizing agricultural production and economic profitability, based on the Green Revolution paradigm and the Food Fordism production model, remains a primary objective of conventional farming to meet global market demands [51,52,53]. Although chemical fertilizers are widely used to enhance crop productivity, nutrient losses make them inefficient [3]. These inefficiencies highlight the need to integrate biofertilizers with conventional farming as a promising strategy to enhance crop yield.

An important finding was the significantly lower accumulation of Fe and Cu in inoculated plants compared to non-inoculated controls. The reported optimal ranges for Fe and Cu in plant tissue are 30–300 mg plant^−1^ and 5–15 mg plant^−1^, respectively [54]. Excessive Fe and Cu accumulation negatively correlated with fruit width, fruit weight, estimated fruit volume (EFV), plant stem width, and chlorophyll (SPAD index), suggesting that biofertilization may mitigate metal stress under conventional input regimes. While Fe is an essential element involved in key biological processes such as cellular respiration, photosynthesis, chlorophyll biosynthesis, metabolism, oxygen transport and balance, DNA synthesis and repair, and metal homeostasis [55], its accumulation has been associated with reduced yields in vegetables crops [4].

Similarly, Cu is a crucial cofactor in several enzymatic systems, including plastocyanin, copper/zinc superoxide dismutase (Cu/ZnSOD), ethylene receptors, and ascorbate oxidase. However, accumulation of Cu can impair plant growth [56]. The ability of inoculated strains to regulate metal dynamics likely reflects their siderophore production. The strains ITTG-R7^T^, ITTG-S70^T^, and ACO-34A are known for their siderophore production, which facilitates metal chelation. Furthermore, their genomes harbor specialized genes involved in Fe regulation. Genes such as *tbp/lbp*, which encode transferrin and lactoferrin proteins, and *hem*, *hmu*, *pig*, *hug*, *chu*, and *hut*, which encode heme proteins, play crucial roles in Fe transport and utilization. Additionally, genes such as *fhu*, *ton*, *exb*, *yus*, *feu*, and *fat* encode siderophores for Fe acquisition [57,58]. For Cu detoxification, genes such as *cusS*, *copS*, and *silS* encode a sensing histidine kinase involved in metal sensing and resistance [59,60], while *mmc0* encodes a multicopper oxidase [61]. Additionally, *copA/ctpA* mediate Cu transport [62] and *rcnA* regulates Ni efflux [63]. Although no visible toxicity symptoms were observed in non-inoculated controls, reduced metal accumulation could represent a critical mechanism that enhances yield stability in agricultural systems. However, further research is needed to elucidate how these native rhizobial strains respond to chemical Fe and Cu fertilization and how this influences gene expression regulation under conventional farming conditions.

Beyond their role in plant growth promotion, the inoculated rhizobial strains coexisted with the native soil bacterial community without causing significant shifts in rhizosphere community structure, reinforcing their role as plant probiotics (PPs). This contrasts with studies reporting that inoculation with plant growth-promoting bacteria (PGPB), different from the *Rhizobium* and *Sinorhizobium* genera, can induce changes in the rhizosphere bacterial community [64,65,66]. This is particularly relevant given the critical role of soil bacterial communities in nutrient cycling, organic matter decomposition, plant health, crop productivity, and soil resilience [67].

Although soil microbial composition is often associated with physicochemical properties such as organic matter and pH [68,69,70], our redundancy analysis (RDA) did not show significant correlations between soil fertility variables, bacterial community structure, and plant performance. Notably, shifts in bulk soil macronutrient and micronutrient concentrations were observed between BS_I and BS_F. However, these changes cannot be directly attributed to the rhizobial inoculation strategy, as the inoculants were applied directly to the tomato microgreens, targeting the rhizosphere rather than the bulk soil. Consequently, the variations in soil nutrient content are more likely explained by a combination of abiotic and biotic factors and crop management practices that influenced soil dynamics. While the native soil bacteriome plays a fundamental role in key ecosystem functions, including nutrient cycling and organic matter decomposition [71], previous studies have documented that tillage and excessive use of chemical fertilizers can disrupt the hydraulic properties and overall quality of agricultural soils. These disruptions include declines in soil organic matter (SOM), pH, and changes in electrical conductivity (EC) [72,73].

Therefore, future research should focus on long-term monitoring of soil properties to better understand the effects of rhizobial inoculants under diverse agronomic management strategies and assessing how these interactions influence soil quality, microbial community, and environmental sustainability, both under conventional and alternative farming systems.

## 5. Conclusions

The findings of this study demonstrate that inoculation with the native rhizobial strains *Sinorizobium mexicanum* ITTG-R7^T^, *Sinorizobium chiapasense* ITTG-S70^T^, and *Rhizobium* sp. ACO-34A positively influenced tomato crop performance under conventional farming. Inoculation with ITTG-R7^T^ significantly increased plant height, stem width, and dry plant biomass, whereas ITTG-S70^T^ enhanced stem width and chlorophyll content. Additionally, plants inoculated with the diazotrophic ACO-34A strain exhibited a significant increase in fruit number, fruit weight, fruit height, fruit width, and estimated fruit volume (EFV) compared to non-inoculated plants. Furthermore, all three elite strains reduced Fe and Cu accumulation in plant tissue. A key aspect of this study is that the application of ITTG-R7^T^, ITTG-S70^T^, and ACO-34A did not significantly alter the native rhizospheric soil bacterial community structure under our experimental conditions. This observed stability is a desirable characteristic for bacterial inoculants in conventional farming systems, as it suggests compatibility with the existing soil microbiome while still providing agronomic benefits.

## Figures and Tables

**Figure 1 microorganisms-13-01904-f001:**
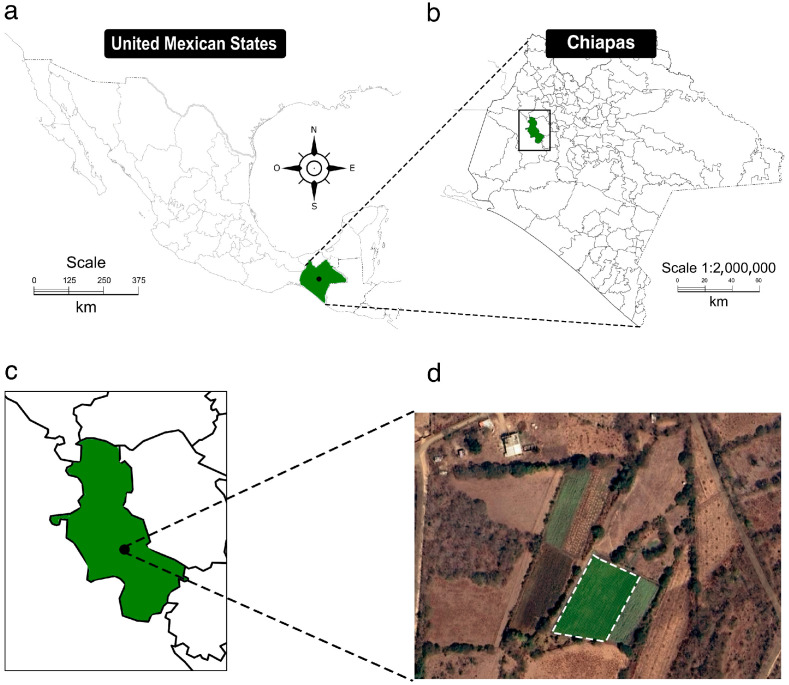
The geographic location of the experimental site. (**a**) The location of the experimental site in Mexico; (**b**) The state of Chiapas and the highlighted location of the “Santa Rosa” area; (**c**) the detailed location of the “Santa Rosa” site, and (**d**) an aerial view of the experimental plot.

**Figure 2 microorganisms-13-01904-f002:**
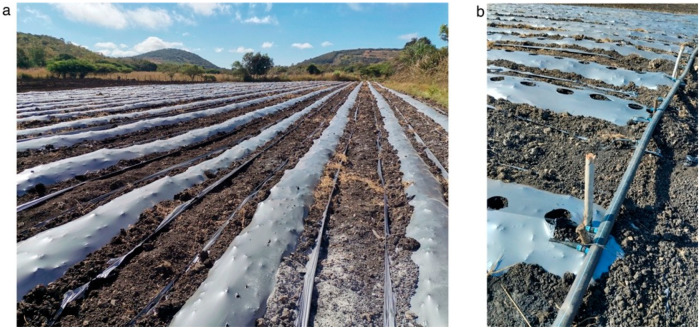
Experimental plot with (**a**) plastic mulch covered ridges and (**b**) micro-sprinkler irrigation system.

**Figure 3 microorganisms-13-01904-f003:**
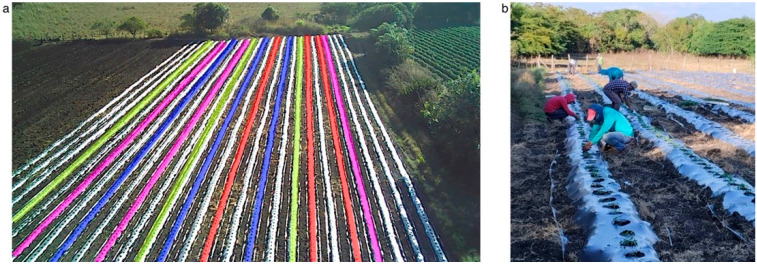
The experimental plot and inoculated tomato microgreens transplantation. (**a**) Treatment distribution in the experimental plot with colors indicating the different treatments: T_1_: non-inoculated (yellow), T_2_: TACO-34A (pink), T_3_: ITTG-R7 (blue), and T_4_: ITTG-S70 (red). White ridges did not form part of the experiment. Three edge ridges on each side were excluded to minimize border effects, and internal white ridges were not assigned to any treatment. (**b**) Transplantation of inoculated tomato microgreens into plastic mulch-covered ridges.

**Figure 4 microorganisms-13-01904-f004:**
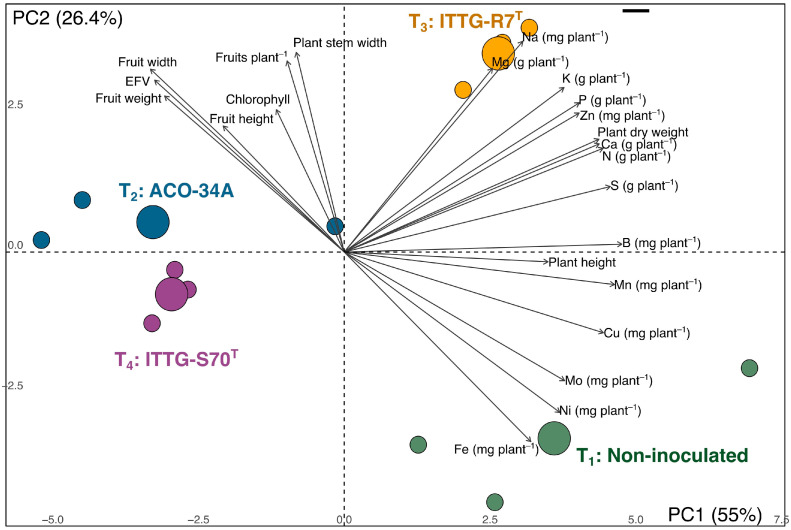
Principal component analysis (PCA) showing the association among morphometric characteristics, fruit yield, and accumulated nutrient content in tomato plant tissue inoculated with native rhizobial bacteria.

**Figure 5 microorganisms-13-01904-f005:**
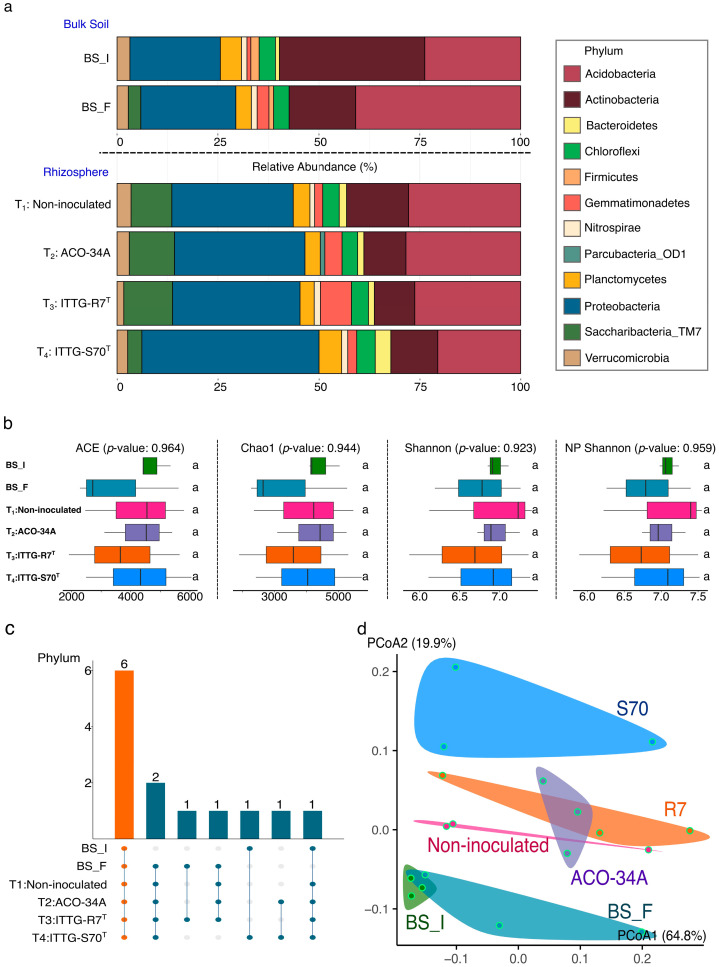
(**a**) The bacterial community structure of bulk soil and rhizospheric soil associated with tomato plants at the phylum level. (**b**) Alpha diversity of bacterial communities based on ACE, Chao1, Shannon, and NP Shannon indices. (**c**) The core bacterial community of agricultural soil at the phylum level. The orange bar and dots represent the number of phyla shared across all samples. Green bars and green dots represent phyla shared among a subset of treatments. Gray dots indicate treatments in which the phylum is not shared. (**d**) Beta diversity of bacterial communities at the species level based on PCoA (Bray–Curtis distance method, PERMANOVA; *p* < 0.05); dots with a green outline represent a biological replicate for each treatment.

**Table 1 microorganisms-13-01904-t001:** Physicochemical characteristics and macro- and micronutrient content in bulk soil.

**Physical characteristics**						
**Sample**	**Texture**	**SP (%)**	**WHC (%)**	**PWP (%)**	**HC (cm h^−1^)**	**BD (g cm^−3^)**
BS_I	Clay	78.47 ± (3.90) A ^Ψ^	42.17 ± (2.16) A	25.07 ± (1.29) A	0.10 ± (0.01) B	1.21 ± (0.05) A
BS_F	Clay	63.36 ± (4.83) A	33.84 ± (2.44) A	20.47 ± (1.74) A	0.69 ± (0.03) A	1.13 ± (0.03) A
*p* < 0.05		0.0631	0.0602	0.0732	0.0005	0.1863
**Chemical characteristics**						
**Sample**	**pH**	**TC (%)**	**EC (dS m^−1^)**	**CEC (meq 100 g^−1^)**	**SOM (%)**	
BS_I	7.96 ± (0.23) A ^Ψ^	26.30 ± (9.01) A	1.18 ± (0.27) A	48.47 ± (1.15) A	5.39 ± (0.16) A	
BS_F	8.67 ± (0.34) A	39.82 ± (1.90) A	1.50 ± (0.04) A	44.0 ± (1.55) A	5.01 ± (0.15) A	
*p* < 0.05	0.1151	0.1419	0.2119	0.0585	0.0768	
**Soil macronutrients (ppm)**						
Sample	**NO_3_**	**P**	**K**	**Ca**	**S**	**Mg**
BS_I ^α^	31.93 ± (1.99) B ^Ψ^	26.10 ± (15.86) A	215.00 ± (9.64) B	8424.33 ± (201.40) A	9.95 ± (1.82) B	708.67 ± (13.58) A
BS_F ^Ω^	38.40 ± (0.94) A	20.24 ± (0.88) A	447.89 ± (35.73) A	7666.33 ± (240.73) A	26.97± (0.98) A	782.67 ± (34.15) A
*p* < 0.05	0.0453	0.5889	0.0055	0.0952	0.0046	0.1134
**Soil micronutrients (ppm)**						
**Sample**	**Na**	**Fe**	**Zn**	**Mn**	**Cu**	**B**
BS_I	24.17 ± (0.21) B ^Ψ^	28.97 ± (0.78) A	1.04 ± (0.01) B	2.85 ± (0.15) A	3.47 ± (0.17) A	1.15 ± (0.04) B
BS_F	27.65 ± (1.01) A	8.51 ± (0.36) B	1.80 ± (0.08) A	3.58 ± (0.18) A	2.17 ± (0.12) B	2.15 ± (0.13) A
*p* < 0.05	0.0377	0.0006	0.0036	0.0588	0.0045	0.0028

^α^ BS_I: bulk soil initial, ^Ω^ BS_F: bulk soil final, SP: saturation point, WHC: water-holding capacity, PWP: permanent wilting point, HC: hydraulic conductivity, BD: bulk density, TC: total carbonates, EC: electrical conductivity, CEC: cation exchange capacity, and SOM: soil organic matter. ^Ψ^ Mean values of three replicates. Means followed by the same letter are non-significant according to the paired *t*-test (*p* < 0.05).

**Table 2 microorganisms-13-01904-t002:** Morphometric traits in tomato plants inoculated with native rhizobial strains.

Treatment	Plant Height (cm)	Plant Stem Width (mm)	Plant Dry Weight (g)	Chlorophyll (SPAD Index)
T_1_: non-inoculated	117.33 ± (3.79) A ^Ψ^	12.45 ± (1.91) C	246.67 ± (68.25) AB	37.09 ± (3.63) D
T_2_: ACO-34 A	85.33 ± (8.39) B	13.89 ± (2.55) B	170.00 ± (52.92) B	39.41 ± (2.90) C
T_3_: ITTG-R7 ^T^	119.33 ± (3.06) A	16.12 ± (3.10) A	306.67 ± (15.28) A	41.26 ± (2.92) B
T_4_: ITTG-S70 ^T^	108.00 ± (15.62) AB	15.41 ± (3.51) A	135.00 ± (13.23) B	43.19 ± (2.39) A
*p*-value	0.0069	0.0000	0.0059	0.0000
HSD ^£^ (*p* < 0.05)	24.0354	0.9425	115.9548	0.9961
CV * (%)	8.51	19.53	20.63	7.44

^£^ HSD: honest significant difference. ^Ψ^ Mean values of three replicates. Means followed by the same letter are non-significant according to Tukey’s test (*p* < 0.05). The standard deviation is given in parentheses. * CV: coefficient of variation. ^T^ Type strain. SPAD: soil plant analysis development index (estimates chlorophyll content in plant leaves).

**Table 3 microorganisms-13-01904-t003:** Fruit size and estimated yield inoculated with native rhizobial strains.

Treatment	Fruits per Plant	Fruit Height (cm)	Fruit Width (mm)	Fruit Weight (g)	EFV (cm^3^)
T_1_: non-inoculated	15.61 ± (7.24) B ^Ψ^	72.37 ± (5.14) B	65.46 ± (6.99) C	155.50 ± (33.67) D	164.19 ± (37.67) C
T_2_: ACO-34A	22.23 ± (5.84) A	77.68 ± (7.40) A	83.75 ± (8.44) A	281.56 ± (86.29) A	291.25 ± (75.63) A
T_3_: ITTG-R7 ^T^	21.91 ± (5.92) A	77.47 ± (5.35) A	79.29 ± (7.50) B	246.92 ± (53.57) B	257.05 ± (53.83) B
T_4_: ITTG-S70 ^T^	21.18 ± (3.81) A	73.25 ± (5.30) B	79.05 ± (6.06) B	218.85 ± (46.40) C	241.62 ± (46.05) B
*p*-value	0.0000	0.0000	0.0000	0.0000	0.0000
HSD ^£^ (*p* < 0.05)	1.9405	2.7588	3.2307	27.4065	25.8819
CV * (%)	28.86	7.81	9.49	25.81	23.17

^£^ HSD: honest significant difference. ^Ψ^ Mean values of three replicates. Means followed by the same letter are non-significant according to Tukey’s test (*p* < 0.05). The standard deviation is given in parentheses. * CV: coefficient of variation. EFV: estimated fruit volume. ^T^ Type strain. Fruits per plant were assessed from 120 plants per treatment, while fruit height, width, and weight were measured from 60 fruits per treatment.

**Table 4 microorganisms-13-01904-t004:** Accumulated nutrient content in dried plant tissue inoculated with native rhizobial strains.

**Macronutrients ^Ω^ (g plant^−1^)**								
**Treatment**	**N**	**P**	**K**	**Ca**	**Mg**	**S**		
T_1_: non-inoculated	4.80 ± (1.28) AB ^Ψ^	0.49 ± (0.14) AB	3.66 ± (0.94) AB	14.56 ± (3.86) A	1.71 ± (0.44) AB	2.75 ± (0.80) A		
T_2_: ACO-34A	3.45 ± (1.06) AB	0.34 ± (0.09) B	3.26 ± (0.99) AB	7.49 ± (2.29) B	2.10 ± (0.62) AB	1.02 ± (0.30) B		
T_3_: ITTG-R7 ^T^	5.67 ± (0.25) A	0.71 ± (0.06) A	4.96 ± (0.29) A	19.12 ± (0.75) A	2.52 ± (0.14) A	3.19 ± (0.15) A		
T_4_: ITTG-S70 ^T^	3.27 ± (0.30) B	0.31 ± (0.04) B	2.92 ± (0.27) B	7.79 ± (0.80) B	1.24 ± (0.14) B	1.28 ± (0.11) B		
*p*-value	0.0253	0.0022	0.0343	0.0007	0.0215	0.0007		
HSD ^£^ (*p* < 0.05)	2.2323	0.2324	1.8542	6.043	1.0318	1.1449		
**Micronutrients ^Ω^ (mg plant^−1^)**								
**Treatment**	**Na**	**Fe**	**Zn**	**Mn**	**Cu**	**B**	**Ni**	**Mo**
T_1_: non-inoculated	133.01 ± (37.08) B ^Ψ^	341.82 ± (93.24) A	45.17 ± (12.19) B	104.04 ± (29.95) A	40.46 ± (11.21) A	15.55 ± (4.23) A	3.73 ± (1.00) A	0.25 ± (0.07) A
T_2_: ACO-34A	135.43 ± (42.76) B	44.06 ± (13.86) B	25.57 ± (7.70) C	21.73 ± (6.62) B	8.30 ± (2.53) C	6.67 ± (2.04) B	0.79 ± (0.25) B	0.13 ± (0.04) B
T_3_: ITTG-R7 ^T^	236.13 ± (11.57) A	40.18 ± (2.13) B	68.82 ± (3.59) A	81.49 ± (4.78) A	24.34 ± (1.27) B	14.37 ± (0.95) A	1.22 ± (0.06) B	0.16 ± (0.01) AB
T_4_: ITTG-S70 ^T^	109.70 ± (10.54) B	41.61 ± (3.91) B	15.71 ± (1.66) C	28.77 ± (3.01) B	8.16 ± (0.75) C	6.51 ± (0.62) B	0.82 ± (0.08) B	0.16 ± (0.02) AB
*p*-value	0.0033	0.0001	0.000	0.0005	0.0004	0.0024	0.0003	0.0382
HSD ^£^ (*p* < 0.05)	76.7702	123.3776	19.542	40.7789	15.1435	6.3124	1.3515	0.1118

^£^ HSD: honest significant difference. ^Ω^ Macronutrient and micronutrient accumulated values were estimated considering only the phyllosphere and excluding fruits. ^Ψ^ Mean values of three replicates. Means followed by the same letter are non-significant according to Tukey’s test (*p* < 0.05). ^T^ Type strain.

## Data Availability

The original data presented in the study are openly available in National Center for Biotechnology Information at https://www.ncbi.nlm.nih.gov/search/all/?term=PRJNA1004673 (accessed on 13 August 2025) as Sequence Read Archive (SRA) under accession number PRJNA1004673.

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
