# Peer review of "Native Rhizobial Inoculation Improves Tomato Yield and Nutrient Uptake While Mitigating Heavy Metal Accumulation in a Conventional Farming System"

_microorganisms, 2025, doi:10.3390/microorganisms13081904_

Round 1

Reviewer 1 Report

Comments and Suggestions for Authors

The manuscript is devoted to the assessment of the effect of rhizobial strains on the growth and yield of tomato plants and the structure of the bacterial community of the rhizosphere in a field experiment. This is a very interesting and actual topic. It is related to the development of traditional agriculture, which still plays a leading role in food production. However, to improve the quality of the manuscript, I propose to make the following changes to it.

  1. Lines 70-71. Examples of biological risks associated with the use of PGPB should be given.
  2. Line 76. Examples of non-legume crops whose growth is enhanced by rhizobia should be given.
  3. Line 97. Specify which plants are planned to be studied.
  4. Line 114. Remove the reference to Fig. 2.
  5. Section 2.3. "A completely randomized design (CRD) was employed, with four treatments, each replicated three times." However, there were 26 ridges in the experiment. This discrepancy should be explained.
  6. Specify the composition of the YEM+Ca2+ medium.
  7. Section 2.6. Briefly describe the Dumas method and AOAC method 2006.03.
  8. Section 2.2. «At the conclusion of the experiment (May 19, 2024)…». Section 2.6. «All measurements and sample collections were performed on March 14, 2023». The start and end dates of the field experiment, the dates of fruit and plant collection for analysis of the parameters should be specified precisely.
  9. Table 4. It is very difficult to understand which row/column the numbers refer to. The font size should be reduced. It is enough to indicate the units of measurement (g plant-1 or mg plant-1) once after the words Macronutrients and Micronutrients.
  10. Why did not the authors analyze the content of micro- and macroelements, as well as vitamins in the fruits of inoculated and non-inoculated plants? In addition to size and weight, nutritional value is a very important indicator for assessing the quality of fruits.
  11. Lines 460-461. The authors did not analyze the length and mass of the plant root systems, so the phrase "The role of IAA is crucial for enhancing root development, which, in turn, increase the plant's capacity to absorb nutrients [48]" has no direct relation to the results obtained.
  12. Lines 497-504. Where did the authors get the information that the genomes of strains ITTG-R7T, ITTG-S70T, and ACO-34A contain the genes indicated on these lines?
  13. section Discussion. The authors analyzed the content of micro- and macroelements in the soil before and after the experiment. However, its change as a result of bacterial inoculation is not discussed. The relationship (or lack thereof) between the content of elements in the soil and plant tissues is also not discussed.

Author Response

Answer to Reviewer 1

Thank you for your valuable comments. Below we detail each suggestion addressed.

The manuscript is devoted to the assessment of the effect of rhizobial strains on the growth and yield of tomato plants and the structure of the bacterial community of the rhizosphere in a field experiment. This is a very interesting and actual topic. It is related to the development of traditional agriculture, which still plays a leading role in food production. However, to improve the quality of the manuscript, I propose to make the following changes to it.

  1. Lines 70-71. Examples of biological risks associated with the use of PGPB should be given.

Thank you for the focused observation. We have added specific examples of bacterial genera associated with respiratory and gastrointestinal infections, which are sometimes used as PGPB in biofertilizer formulations

  1. Line 76. Examples of non-legume crops whose growth is enhanced by rhizobia should be given.

Thank you for your suggestion. We have now included specific examples of non-legume crops.

  1. Line 97. Specify which plants are planned to be studied.

Thank you for your observation. We have now clarified in the text that tomato (Solanum lycopersicum) was the plant species studied.

  1. Line 114. Remove the reference to Fig. 2.

We appreciate your observation. The suggested modification has been made in the manuscript.

  1. Section 2.3. "A completely randomized design (CRD) was employed, with four treatments, each replicated three times." However, there were 26 ridges in the experiment. This discrepancy should be explained.

We thank the reviewer for the accurate observation. Additionally, we identified an error in figure numbering within Section 2.3, which has now been corrected. Figure 3a shows the randomized distribution of the four evaluated treatments. Although the farmer initially established 26 ridges, 6 of them (3 on each side) were excluded to minimize border effects. Thus, only 20 ridges remained available, from which 12 ridges (4 treatments × 3 replicates) were used for the experimental design. This is not a discrepancy in the number of experimental units, but rather a result of working within a real-world production setting, where the experimental layout was adapted to field conditions established by the farmer.

  1. Specify the composition of the YEM+Ca2+ medium.

Thank you for your observation. The composition of the YEM+Ca² medium has been added in the Methods section as requested.

  1. Section 2.6. Briefly describe the Dumas method and AOAC method 2006.03.

We appreciate the reviewer’s suggestion. We have now included concise technical descriptions of both methods.

  1. Section 2.2. «At the conclusion of the experiment (May 19, 2024)…». Section 2.6. «All measurements and sample collections were performed on March 14, 2023». The start and end dates of the field experiment, the dates of fruit and plant collection for analysis of the parameters should be specified precisely.

Thank you for the observation. The experimental phase began with the collection of bulk soil samples on December 22, 2023. Inoculation and transplantation of tomato plants were carried out on December 30, 2023, as described in Section 2.5. Morphometric measurements and fruit harvesting were conducted on March 14, 2024, and final bulk soil samples were collected on May 19, 2024. Accordingly, we corrected a typographical error in Section 2.6, where the original date stated as March 14, 2023, has been corrected to 2024.

  1. Table 4. It is very difficult to understand which row/column the numbers refer to. The font size should be reduced. It is enough to indicate the units of measurement (g plant-1 or mg plant-1) once after the words Macronutrients and Micronutrients.

Thank you for the suggestion. Table 4 has been reformatted to improve clarity and alignment between rows and columns. The font size was slightly reduced, and the units of measurement are now indicated only once after the labels Macronutrients and Micronutrients, as recommended.

  1. Why did not the authors analyze the content of micro- and macroelements, as well as vitamins in the fruits of inoculated and non-inoculated plants? In addition to size and weight, nutritional value is a very important indicator for assessing the quality of fruits.

We appreciate this valuable suggestion. The present study focused on assessing the agronomic performance and nutrient uptake capacity of tomato plants through analysis of macro- and micronutrients in foliar tissue, which is a widely accepted method to evaluate the effectiveness of plant probiotic inoculants. While we agree that nutrient and vitamin content in fruits is essential for evaluating fruit quality, it was beyond the scope of this work. Nonetheless, we consider it a highly relevant direction for future research.

  1. Lines 460-461. The authors did not analyze the length and mass of the plant root systems, so the phrase "The role of IAA is crucial for enhancing root development, which, in turn, increase the plant's capacity to absorb nutrients [48]" has no direct relation to the results obtained.

Thank you for the observation. We have revised the sentence to avoid any unsupported inference and now refer only to the general association between IAA-producing rhizobia and improved nutrient absorption, based on previous literature.

  1. Lines 497-504. Where did the authors get the information that the genomes of strains ITTG-R7T, ITTG-S70T, and ACO-34A contain the genes indicated on these lines?

Thank you for this observation. The presence of the genes mentioned in the manuscript was confirmed through manual searches performed on the assembled genomes of strains ITTG-R7T, ITTG-S70T, and ACO-34A. The search was guided by previously published literature, already cited in the manuscript, that associates these genes with nutrient metabolism in bacteria. Although a full genome annotation was not conducted as part of this study, the targeted identification of relevant genes supports the discussion of potential mechanisms involved in nutrient uptake observed in plant tissue.

  1. section Discussion. The authors analyzed the content of micro- and macroelements in the soil before and after the experiment. However, its change as a result of bacterial inoculation is not discussed. The relationship (or lack thereof) between the content of elements in the soil and plant tissues is also not discussed.

Thank you for your valuable observation. The discussion section has been revised accordingly to incorporate a more detailed interpretation of the relationships between soil characteristics and macro- and micronutrient content.

-Figures and tables can be improved

Thank you for your observation. We have carefully reviewed all figures and tables in the manuscript and have improved their formatting, and overall clarity to enhance readability and visual impact.

Reviewer 2 Report

Comments and Suggestions for Authors

The manuscript presents interesting research results on probiotic bacteria in tomato cultivation. It was found that inoculation with ITTG-R7T significantly improved plant height, stem width and dry weight, while ITTG-S70T increased stem width and chlorophyll content. Inoculation with ACO-34A significantly increased the number of fruits, their size and yield parameters. How can agricultural producers use these results? What are the risks? What are the disadvantages of this technology? What are the costs? What needs to be investigated now?

Comments

Materials and Methods

Please describe the weather conditions during the study period based on long-term data.

Sodium (Na) is not a microelement but a beneficial element.

Please provide fertiliser doses in elemental form, not oxide form.

Results

The figures are clear and understandable.

Tables 1 and 4.

It is not a microelement but a beneficial element.

Discussion

It is well conducted.

Conclusions

They are short and understandable.

References

A large number of publications (69). Please limit yourself to the most recent publications (no more than 10 years old). Some names are written with a lowercase letter.

Author Response

Reviewer 2

Thank you for your valuable comments. Below we detail each suggestion that was addressed.

The manuscript presents interesting research results on probiotic bacteria in tomato cultivation. It was found that inoculation with ITTG-R7T significantly improved plant height, stem width and dry weight, while ITTG-S70T increased stem width and chlorophyll content.

Inoculation with ACO-34A significantly increased the number of fruits, their size and yield parameters. How can agricultural producers use these results?, What are the risks?, What are the disadvantages of this technology?, What are the costs?, What needs to be investigated now?.

We sincerely thank you for your interest, review, and valuable feedback on our work. In response to your observations, we believe this study provides agricultural producers, with proper technical guidance, more effective strategies for the use of biofertilizers formulated with native rhizobial genetic resources. It also encourages the adoption of agronomic practices that incorporate soil fertility characterization as a key tool to improve crop yields.

Regarding potential risks, the use of bacterial genera such as Rhizobium and Sinorhizobium poses no threat to human, animal, or plant health, nor to ecosystem balance. The strains used in this study are reference strains, genomically and functionally characterized. However, a major risk in the biofertilizer sector is the lack of regulation concerning the microbial content of commercial products, many of which do not guarantee the absence of pathogenic microorganisms. Therefore, it is essential that biofertilizers be formulated with reference strains belonging to genera not associated with pathogenicity.

Among the disadvantages of this technology, it is worth noting that biofertilizers generally have a shorter shelf life compared to chemical fertilizers. Moreover, their effectiveness strongly depends on correct application; if misapplied, they may fail to produce the desired outcomes. In addition, producing a high-quality biofertilizer is more expensive than producing conventional fertilizers. For instance, while one kilogram of urea may cost around one dollar, one kilogram or liter of a biofertilizer formulated with a reference strain can range from 30 to 100 dollars (commercial price). Nonetheless, well-formulated biofertilizers are often applied at low doses (e.g., one liter per hectare), whereas chemical fertilizers—despite their lower cost per kilogram—are used in much larger quantities, often resulting in expenditures exceeding 250 dollars per hectare. Importantly, the environmental impact of excessive chemical fertilizer use is considerably higher than that of biofertilizers.

Therefore, it is necessary to continue validating the performance of biofertilizers under real field conditions and across diverse agricultural management systems (both conventional and alternative) in economically important crops. Future research should evaluate parameters such as inoculant concentration, application volume and frequency, and soil-type compatibility to ensure efficacy and long-term sustainability.

Comments

Materials and Methods

Please describe the weather conditions during the study period based on long-term data.

Thank you for your comment. The weather conditions during the study period, including climate classification, average annual temperature, and total precipitation, are already described in the Materials and Methods section, 2.1. Field site description.

Sodium (Na) is not a microelement but a beneficial element.

Thank you for your valuable observation. We agree with your comment—although sodium (Na) is not formally classified as a micronutrient, it was included in the comparative nutrient analysis due to its presence in plant tissue and its variation across treatments.

Please provide fertilizer doses in elemental form, not oxide form.

Thank you for your observation. We acknowledge that expressing fertilizer doses in elemental form is standard practice. However, in this study, we chose to report the fertilizers in their oxide or salt forms for two main reasons:

(1) To accurately reflect the specific commercial formulations used during field management, which may help other producers identify and relate to the actual products applied under the described agronomic conditions; and

(2) To facilitate the potential replication of the fertilization strategy by other research groups or agricultural stakeholders using the same formulations.

Nonetheless, if the editorial team considers it necessary, we are open to including the equivalent elemental doses as supplementary information for clarity and transparency.

Results

The figures are clear and understandable. Tables 1 and 4. It is not a microelement but a beneficial element.

Thanks for the feedback about the figures and tables. We have reviewed and corrected some details to improve understanding.

Discussion. It is well conducted.

Conclusions. They are short and understandable.

Thanks for the feedback about the discussion and conclusions.

References

A large number of publications (69). Please limit yourself to the most recent publications (no more than 10 years old). Some names are written with a lowercase letter.

We sincerely thank the reviewer for this valuable observation regarding the number and formatting of the references. In response, we conducted a comprehensive review of the reference list.